# Erosion Failure of Slope in a Dump with Ground Fissure under Heavy Rain

Yexin Li [1], Gang Lv [2,*], Daohan Wang [2], Wenxuan Su [1] and Zhongping Wei [3]

[1] School of Architecture and Civil Engineering, Shenyang University of Technology, Shenyang 110870, China
[2] College of Environmental Science and Engineering, Liaoning Technical University, Fuxin 123000, China
[3] Liaoning Academy of Forestry Science, Shenyang 110032, China
* Correspondence: 18524182288@163.com

**Abstract:** The dump, with the compact rock platform and high and steep loose slope that is formed during coal mining, is the most serious area of soil erosion in a surface coal mine. Ground fissures are a typical geological hazard in coal mining areas. However, the effect of ground fissures on soil erosion remains unclear. Rainfall experiments were conducted to determine the varying characteristics of wetting front, runoff and sediment production, and soil denudation rate, as well as the effects of ground fissures on these factors in a platform-slope system of a dump. Ground fissures could significantly enhance wetting front and soil erosion. Rill erosion was formed as the rainfall and runoff flushed the soil, which eventually developed into erosion gullies. Erosion failure modes with platform-slope systems in the dump could be divided into the surface erosion stage, fissure deformation stage, rill erosion stage, fissure collapse-rapid increase stage, and stable stage. Runoff power and flow shear stress had the greater influence on soil denudation rate, which indicated that erosion energy of concentrated flow had important influence on soil erosion. Moreover, shallow mudflow induced by rainfall was one of the forms of soil slope instability; it occurred in a short time with great soil erosion. Soil erosion in the dump with ground fissures was mainly shallow mudflow and rill erosion, resulting from the combined effect of hydraulic erosion and gravity erosion.

**Keywords:** soil erosion; shallow mudflow; ground fissure; coal mining area; rainfall

## 1. Introduction

Coal is an irreplaceable stable main energy source in the short term in China [1,2]. In 2018, the total national coal consumption was 2.74 billion tons of standard coal, providing energy security for China's development [3]. However, the mining of coal resources can not only bring about rapid economic development, but also cause serious ecological and environmental problems, among which open-pit mining is the most serious [4,5]. During the 55 years from 1950 to 2005, about 10,869 geological disasters occurred in China's mining developments, with 4779 deaths and direct economic losses of about RMB 17.458 billion [6]. In particular, mine spoils are an important driver of environmental damage land degradation [7–9]. Hence, it is necessary to study the geological disasters in coal mining areas.

Most of the large coal mines are located in the vulnerable environments of northwestern China [10]. As a typical geomorphic unit of the mining area, the dump has loose slopes with steep slopes and long slopes, a platform with rock and soil compaction, complex material composition, developed porosity, uneven subsidence, etc. [11]. The dump is the most serious area of soil erosion in a surface coal mine [10], with multiple soil erosion types, such as splash erosion, surface erosion, rill erosion, collapse, landslide, and debris flow, and so on [12–14]. The process of soil erosion in the dump is special and complex, and the magnitude is extremely serious. The erosion rate in a dump is 43.6–239.2 times that of abandoned land [15]. Therefore, it is necessary to study soil erosion of the dump in a surface coal mine.

Ground fissures (GF) are a typical geological hazard in coal mining areas. They occur in large numbers with extensive distribution, causing severe damage and seriously affecting the ecological security of mining areas [16–18]. They not only cause the decrease of land productivity and serious water pollution, but are also a threat to mine electrical and mechanical equipment and production personnel safety [19,20]. Ground fissures have relatively strong spatial variability, yet clear self-similarity. Ground fissures change the movement path of surface runoff, so that surface runoff and rainfall directly move from the fissures to the interior of the soil, reducing the stability of the soil [21]. Relevant research shows that the appearance of ground fissures improves the soil infiltration capacity and is the main reason for slope deformation and instability [22,23]. Hence, the occurrence of ground fissures greatly affects the hydrological cycle processes such as surface runoff, infiltration, and evaporation, and also increases the possibility of soil and water loss disasters such as collapse, landslide, and debris flow in the dump [24]. Therefore, understanding the laws of water movement with ground fissures in a dump response to soil erosion is essential.

Research on soil erosion in a surface coal mine dump includes soil erosion mechanisms and characteristics [25], influencing factors [7], and erosion prediction [15]. Of course, some researchers have focused on the role and contribution of the platform of the dump in soil erosion [26,27]. However, few studies have been carried out regarding the soil erosion on platform and slope in a dump. In China, studies related to the source of erosion sediment of the dump model [28], and the soil erosion characteristics of the dump under different rainfall intensity [29] are rare. Platform catchment is a key link that affects the process and amount of soil erosion in the dump. The surface runoff collected by the platform in the dump provides erosion power, and the preferential flow caused by ground fissures promotes soil erosion and slope instability. In addition, the slope of the dump was the main source of soil erosion. Therefore, it is necessary to take the platform and slope of the dump as a system to reveal the erosion failure characteristics.

Therefore, taking a platform–slope system of the dump as an example, the soil erosion process and erosion failure characteristics were studied by a simulated rainfall test. The goals of this study were to analyze: (1) the wetting front depth with time, (2) the process of soil erosion and effect of ground fissures on erosion failure, and (3) the relation between the soil denudation rate and hydraulic parameters in a surface coal mine dump. These results are of significance for controlling soil erosion in a dump.

## 2. Materials and Methods

### 2.1. Study Site

The study area is in the south dump of Shenglidong Open-pit Coal Mine No. 2 of Datang International Power Generation Company in Xilinhot city, Xilingol league, Inner Mongolia (Figure 1). The local climate is arid and semiarid in the middle temperate zone. The average annual temperature is 1.7 °C, and the average annual precipitation is 284.74 mm. Precipitation mainly occurs in June–August, accounting for more than 71% of the annual rainfall. The annual average evaporation is 1794.6 mm, and the annual average wind speed is 3.4 m/s. The climate conditions are mainly derived from the China Meteorological science data sharing service platform, with an average value of 30 years. The soil is typical chestnut soil. To restore the vegetation of the dump as soon as possible, soil covering measures were applied for the platform and slope reclamation (the soil is sandy loam).

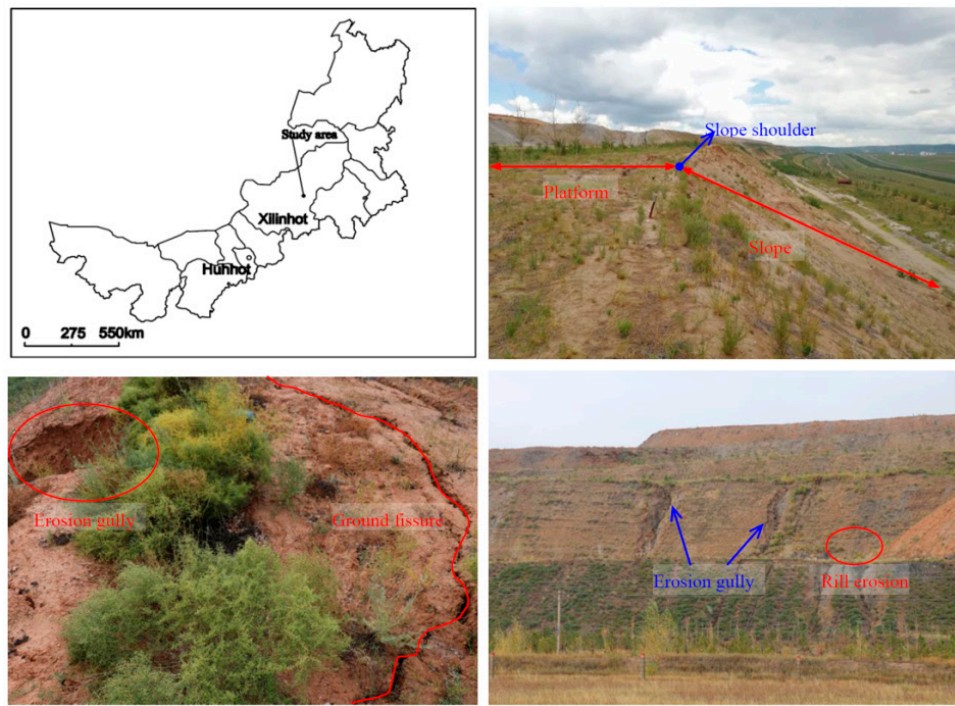

**Figure 1.** Location of the study area.

## 2.2. Experimental Design

When conducting indoor simulation research, the similarity principle was usually used to determine the geometric size of the dump, and the similar proportional constant between the actual dump in the field and the simulated dump in the room was used as the basis for the experimental design. The similarity coefficient of this study was 50. The dump slope was made of sandy loam soil and coal gangue with a thickness of 50 cm and the thickness of each layer was 10 cm. The lower layer was coal gangue with a thickness of 20 cm, a particle size of 10–30 mm, and a soil bulk density of 1.7 g/cm$^3$. The upper layer was sandy loam with a thickness of 30 cm, a particle size of 0–10 mm, and a soil bulk density of 1.35 g/cm$^3$. According to the rainstorm occurrence frequency based on the rainfall data, the rainfall intensity was set to 1.5 mm/min. According to the survey results of the dump, the designed ground fissure depths were 5, 10, 15, and 20 cm and these tests were marked GF1, GF2, GF3, and GF4, respectively (Table 1).

**Table 1.** Each conditions with different ground fissure.

| Test | Dimension (cm) | Ground Fissure Depth (cm) | Ground Fissure Volume (cm$^3$) | Platform Catchment Area (cm$^2$) |
|------|---------------|---------------------------|-------------------------------|----------------------------------|
| GF1 | 110 × 50 × 50 | 5 | 375 | 2000 |
| GF2 | 110 × 50 × 50 | 10 | 750 | 2000 |
| GF3 | 110 × 50 × 50 | 15 | 1250 | 2000 |
| GF4 | 110 × 50 × 50 | 20 | 1500 | 2000 |

The experimental facilities are shown in Figure 2. The artificial rainfall simulator was portable with 10 water sprayers and it was same as that of Lv et al. [30]. Each water sprayer could be controlled independently, and the number of water sprayers to be used depended on the specific experimental conditions. The thickness of glass flume was 10 mm and the length, width, and height were 110, 50, and 60 cm, respectively. The excess infiltration water was removed by 4 rows and 4 columns of circular holes. The equivalent model of ground fissures was determined by the survey results and related research [31], which was made of a thin rigid metal plate. The length of the equivalent model was 30 cm, and the height was 5, 10, 15, or 20 cm (Table 1). During the filling process, the equivalent model

of the ground fissure was embedded and taken out after 48 h. The outlet was set at the front edge of the glass flume. After the occurrence of slope runoff, the time of runoff was recorded. Then, the runoff sediment samples were collected every 3 min and the rainfall lasted for 60 min. The number of erosion gullies were calculated after rainfall, and their length, width, depth were measured by steel ruler.

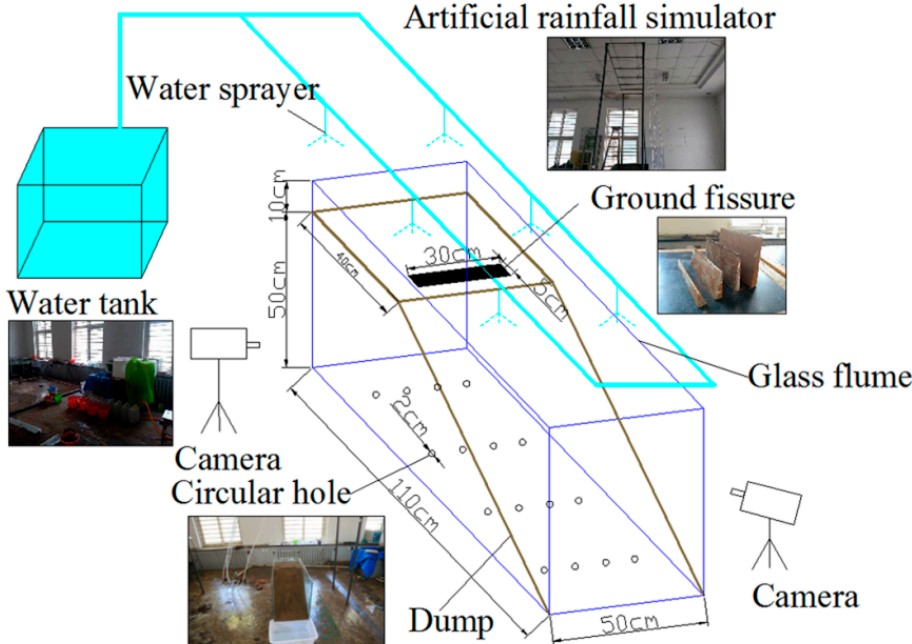

**Figure 2.** Schematic of the experimental setup.

*2.3. Data Analysis*

　　The soil denudation rate and hydraulic parameters were determined from by Peng et al. [32] and Zhang et al. [33]. The Grey Relational Analysis was used to analyze their relationship [34]. The Pearson correlation coefficients and regression equation were used by Origin 9.1 and SPSS 17.0 software.

## 3. Results

*3.1. Wetting Front*

　　Figure 3 indicates the variation characteristics of wetting fronts with different ground fissures under rainfall conditions. As shown in Figure 3, the wetting front increased continuously. The depth of the wetting front for GF1, GF2, GF3, and GF4 were 4.0, 4.3, 4.5, and 4.5 cm at 5 min, respectively, and then increased in different degrees. At the end of the rainfall test, the depth of the wetting front for GF1, GF2, GF3, and GF4 were 14.5, 17.8, 22.7, and 30.5 cm, respectively. The relationship between the wetting front and time with different fissure depths were as follows:

$$5 \text{ cm depth (GF1) } y = 1.619x^{0.534} \ (R^2 = 0.996; p = 0.000; n = 12)$$

$$10 \text{ cm depth (GF2) } y = 1.706x^{0.578} \ (R^2 = 0.997; p = 0.000; n = 12)$$

$$15 \text{ cm depth (GF3) } y = 2.45x^{0.553} \ (R^2 = 0.975; p = 0.000; n = 12)$$

$$20 \text{ cm depth (GF4) } y = 1.919x^{0.69} \ (R^2 = 0.988; p = 0.000; n = 12)$$

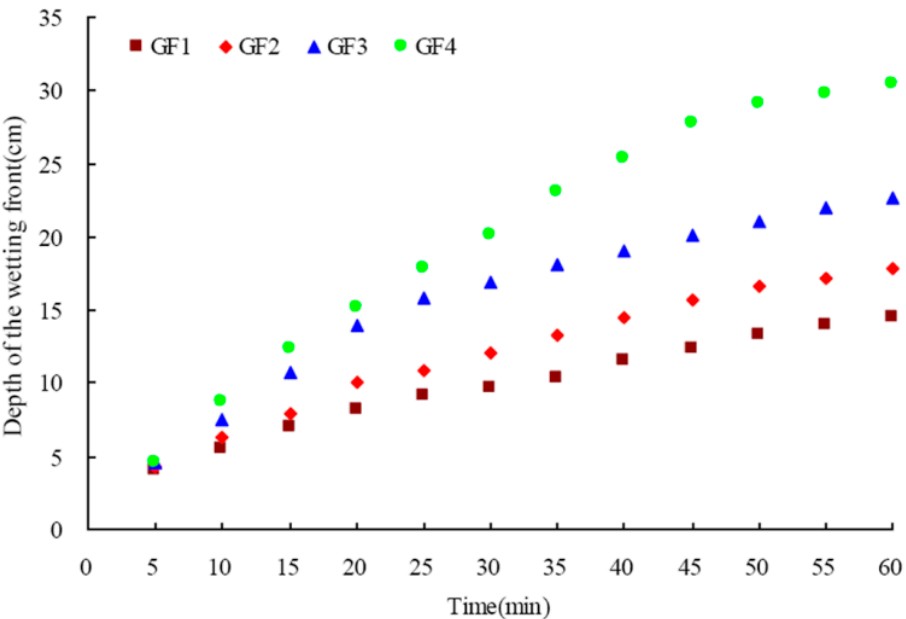

**Figure 3.** Variation characteristics of wetting fronts.

### 3.2. Soil Erosion Process

The soil on the slope was changed from dry to wet, and splash erosion and surface erosion occurred successively. The rill erosion developed into gully erosion by rainfall and runoff (Figure 4). The soil water content was low at first, and then increased, and the thin layer flow was formed. It provided a foundation for the occurrence of surface erosion. As the same time, a large amount of rainwater and surface runoff flowed into the ground fissure, and then moved to the deep soil. With the water volume increasing, the water level of the ground fissure increased, and the soil stability decreased, which caused soil collapse in the ground fissure. When the runoff on the slope had a certain erosive ability, rill erosion occurred, and it first appeared at the bottom of the slope. Next, rill erosion moved upwards, and multiple rills on the slope were formed. As the rainfall time increased, under the combined action of water pressure in the ground fissure and rill erosion, the soil at the slope shoulder was washed away. The result was the rapid increase of runoff and erosion.

Comparing the soil erosion of dump at different ground fissures depth, it could be seen that the greater the ground fissure depth, the more serious the soil erosion. It was observed that the number of erosion gullies with GF1, GF2, GF3, and GF4 were 5, 4, 3, and 3, respectively. For GF1, the lengths of all the erosion gullies were 36.4, 43.2, 45.4, 50.3, and 60.5 cm, the corresponding widths and depths were 5.67, 6.31, 5, 4.65, and 5 cm, and 3, 2, 3, 4.63, and 6, respectively. The soil erosion of GF4 was faster and more serious than that of GF1. There were 3 erosion gullies for GF4 with lengths of 50.40, 55.60, and 86.10 cm, widths of 12.32, 9, and 26 cm, and depths of 9.67, 7.38, and 14 cm, respectively.

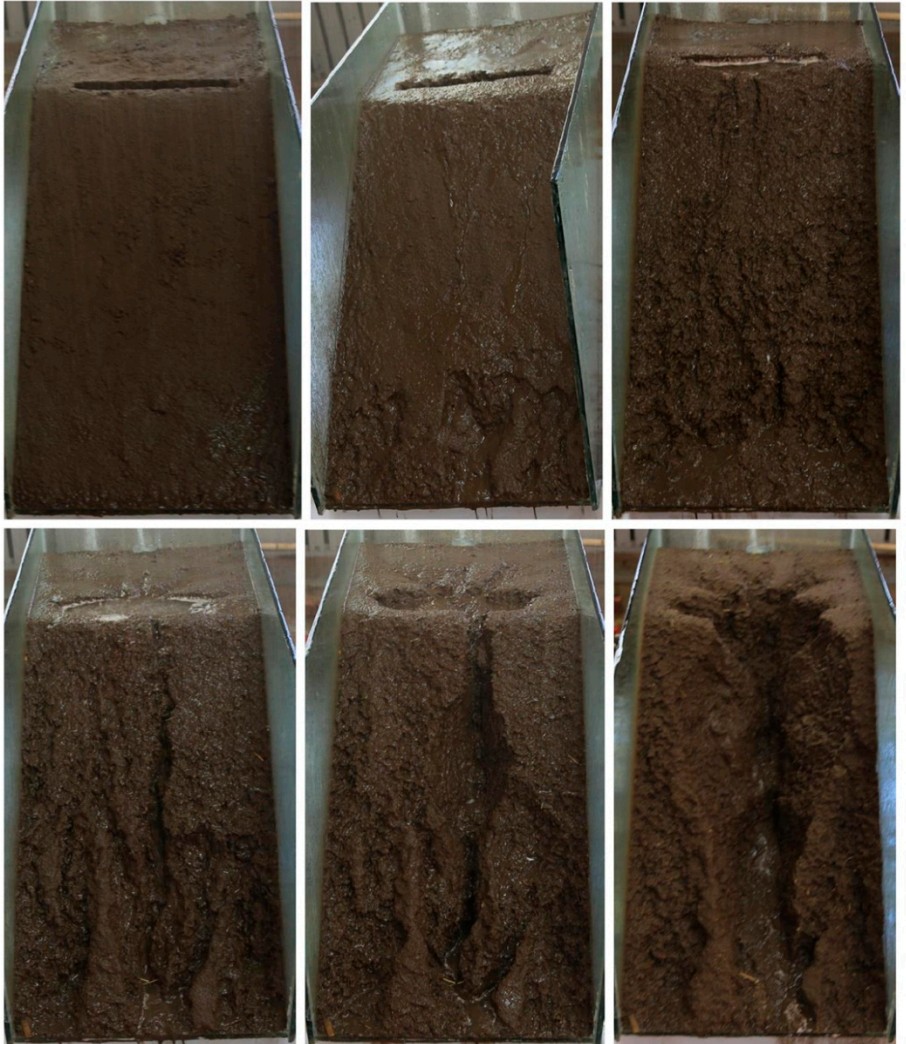

**Figure 4.** Process of soil erosion under rainfall.

### 3.3. Runoff and Sediment

As shown in Figure 5, for GF1, with the rainfall time increasing, the runoff rate increased and then stabilized. The runoff rate had a similar variation law for GF2, GF3, and GF4, but there was a special point for GF2. When the rainfall time increased from 3 to 6 min, the runoff rate increased by 4.13 times. The runoff rate at 6 min was 2.36 L/min, which was 2.41 times the average. This was the main reason for the fluctuation of the runoff rate with time. The average runoff rates for GF1, GF2, GF3, and GF4 were 0.39, 0.98, 0.59, and 1.27 L/min, respectively, which demonstrates increases of 150.81%, 50.54%, and 224.53% between the rates. The cumulative runoff with different fissure depths were 23.41, 58.7, 35.24, and 75.95 L, which increased significantly with time. The relationship between cumulative runoff and time with different fissure depths were as follows:

$$\text{5 cm depth (GF1) } y = 0.131x^{1.268} \ (R^2 = 0.999; p = 0.000; n = 20)$$

$$\text{10 cm depth (GF2) } y = 2.443x^{0.772} \ (R^2 = 0.993; p = 0.000; n = 20)$$

$$\text{15 cm depth (GF3) } y = 0.261x^{1.202} \ (R^2 = 0.998; p = 0.000; n = 20)$$

$$\text{20 cm depth (GF4) } y = 0.356x^{1.302} \ (R^2 = 0.998; p = 0.000; n = 20)$$

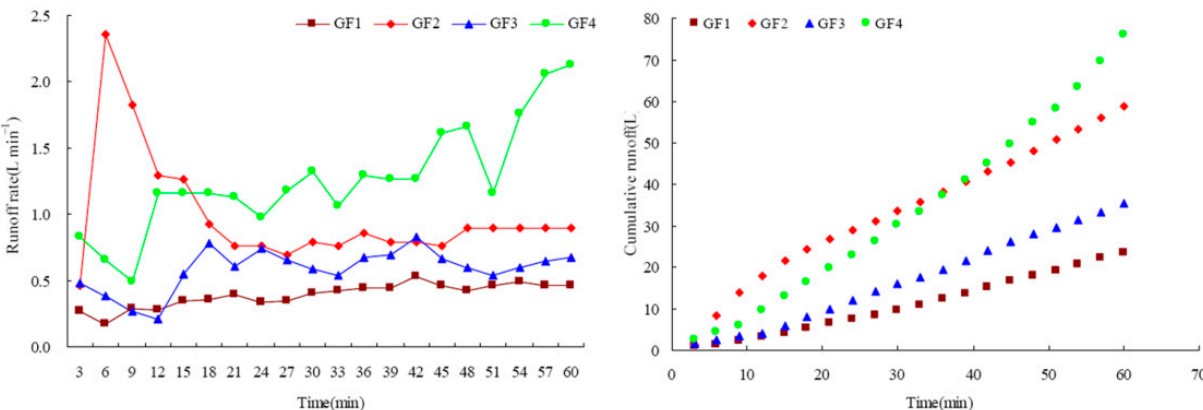

**Figure 5.** Variation of runoff rate and cumulative runoff with time.

As shown in Figure 6, it could be seen that the erosion rate presented successive fluctuations with multiple peaks and valleys. At the beginning of the test, the runoff erosion capacity was low, not enough to wash out soil. Then the erosion rate increased and fluctuated because of the rill erosion. Under the action of runoff, the erosion pattern on the slope changed by the merging and bifurcation of the rills, and then the erosion rate also changed. For GF1, the erosion rate varied from 46.92 to 82.18 g/min, and its value was lower than that of GF2, GF3, and GF4. The erosion rates varied from 135.62 to 552.62 g/min and from 198.65 to 863.74 g/min for GF3 and GF4. The average erosion rates for GF1, GF2, GF3, and GF4 were 63.20, 327.66, 383.45, and 576.16 g/min, respectively, which shows increases of 418.49%, 506.78%, and 811.72%. The cumulative sediment with different fissure depths were 3791.71, 19659.47, 23007.26, and 34569.88 g, which increased significantly with time. The relationship between cumulative sediment and time with different fissure depths were as follows:

$$5 \text{ cm depth (GF1) } y = 79.489x^{0.947} \ (R^2 = 0.999; p = 0.000; n = 20)$$

$$10 \text{ cm depth (GF2) } y = 5634.448x^{0.325} \ (R^2 = 0.739; p = 0.000; n = 20)$$

$$15 \text{ cm depth (GF3) } y = 214.592x^{1.148} \ (R^2 = 0.996; p = 0.000; n = 20)$$

$$20 \text{ cm depth (GF4) } y = 215.182x^{1.234} \ (R^2 = 0.996; p = 0.000; n = 20)$$

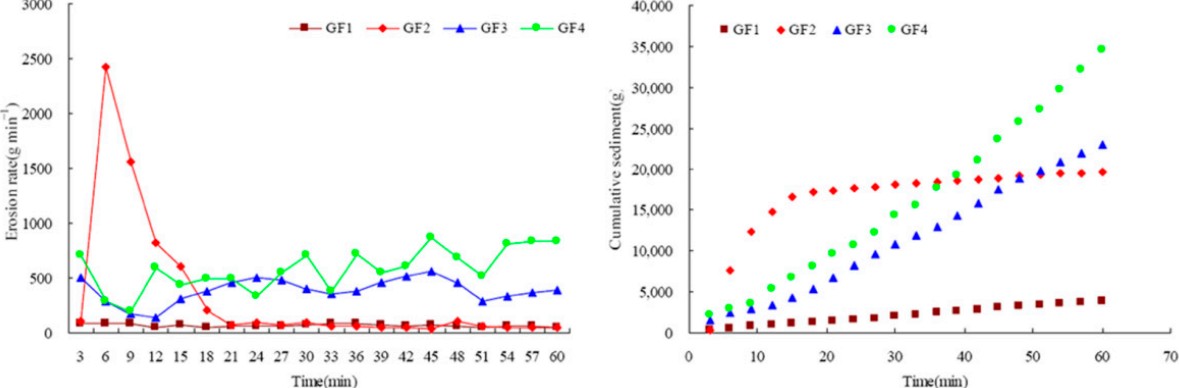

**Figure 6.** Variation of erosion rate and cumulative sediment with time.

### 3.4. Soil Denudation Rate

It could be seen that variation in the trend of soil denudation rate was similar to that of erosion rates (Figure 7). In particular, the fluctuation of GF4 was the most violent. For GF1, the soil denudation rate varied from 121.99 to 249.41 g/m²/min, and its value was lower than that of GF2, GF3, and GF4. The maximum soil denudation rate for GF2 was as high

as 5633.11 g/m$^2$/min at 6 min. Then, soil denudation rates for 9, 12, 15, and 18 min were 3774.78, 2066.25, 1569.78, and 529.63 g/m$^2$/min, respectively. For GF3, the maximum soil denudation rate was 2471.45, and the minimum was 328.55. For GF4, the soil denudation rate continued to increase, with a maximum of 5352.59. The average soil denudation rates for GF1, GF2, GF3, and GF4 were 191.26, 842.1, 1369.43, and 2367.3 g/m$^2$/min, respectively, which show increases of 340.3%, 616%, and 1137.73%.

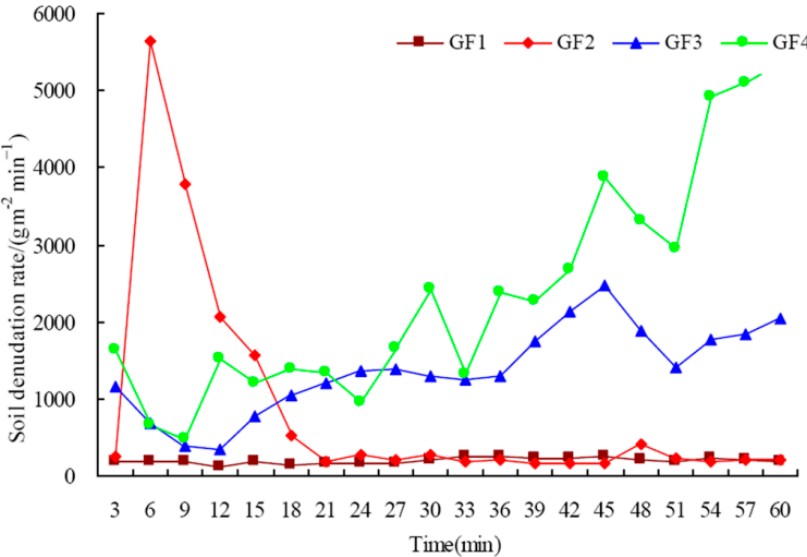

**Figure 7.** Variation of soil denudation rate with time.

Figure 8 and Table 2 indicate the relationship between soil denudation rates and hydraulic parameters. It could be seen that the soil denudation rate showed an increasing trend with increasing flow velocity, flow shear stress, runoff power, Reynolds number, and Froude number, while the Darcy–Weisbach roughness coefficient showed a decreasing trend. Table 2 indicates that soil denudation rates with different fissure depths had different correlations with hydrodynamic parameters. For GF1, the soil denudation rate significantly increased with Reynolds number (Pearson correlation coefficients $R = 0.496$; $p = 0.026 < 0.05$) and Froude number ($R = 0.631$; $p = 0.003 < 0.01$), and decreased with Darcy–Weisbach roughness coefficient ($R = 0.680$; $p = 0.001 < 0.01$) (Table 3). For GF2, the soil denudation rate significantly increased with flow velocity ($R = 0.792$; $p = 0.000$), flow shear stress ($R = 0.984$; $p = 0.000$), runoff power ($R = 0.988$; $p = 0.000$), Reynolds number ($R = 0.447$; $p = 0.048 < 0.05$) and Froude number ($R = 0.596$; $p = 0.006 < 0.01$), and decreased with Darcy–Weisbach roughness ($R = 0.485$; $p = 0.03 < 0.05$). For GF3, the soil denudation rate significantly increased with flow velocity ($R = 0.639$; $p = 0.002 < 0.01$), and decreased with Darcy–Weisbach roughness ($R = 0.56$; $p = 0.01 < 0.05$). For GF4, the soil denudation rate significantly increased with flow velocity ($R = 0.774$; $p = 0.000$), runoff power ($R = 0.654$; $p = 0.002 < 0.01$), and Reynolds number ($R = 0.701$; $p = 0.001 < 0.01$). Moreover, it could be seen that a significant correlation was found between soil denudation rate and hydraulic parameters on all the ground fissures, for which Pearson correlation coefficients ranged in such order as runoff power (0.771) > flow velocity (0.764) > Reynolds number (0.709) > flow shear stress (0.659) > Froude number (0.327) > Darcy–Weisbach roughness coefficient (0.326). Comparing different hydraulic parameters, it could be seen that the correlation between soil denudation rates and runoff power was the highest, and the correlation with the Darcy–Weisbach roughness coefficient was the lowest.

As shown in Table 3, the hydraulic parameters were ranked in the order of runoff power (0.866) > flow shear stress (0.851) > Reynolds number (0.801) > flow velocity (0.781) > Froude number (0.763) > Darcy–Weisbach roughness coefficient (0.716). Among these hydraulic parameters, runoff power and flow shear stress were of great influence on soil denudation rate, which indicated that erosion energy of concentrated flow had impor-

tant influence on soil erosion. The Darcy–Weisbach roughness coefficient had the weakest correlation with soil detachment rate, which was essentially identical to that obtained using the Pearson correlation coefficients (Table 2).

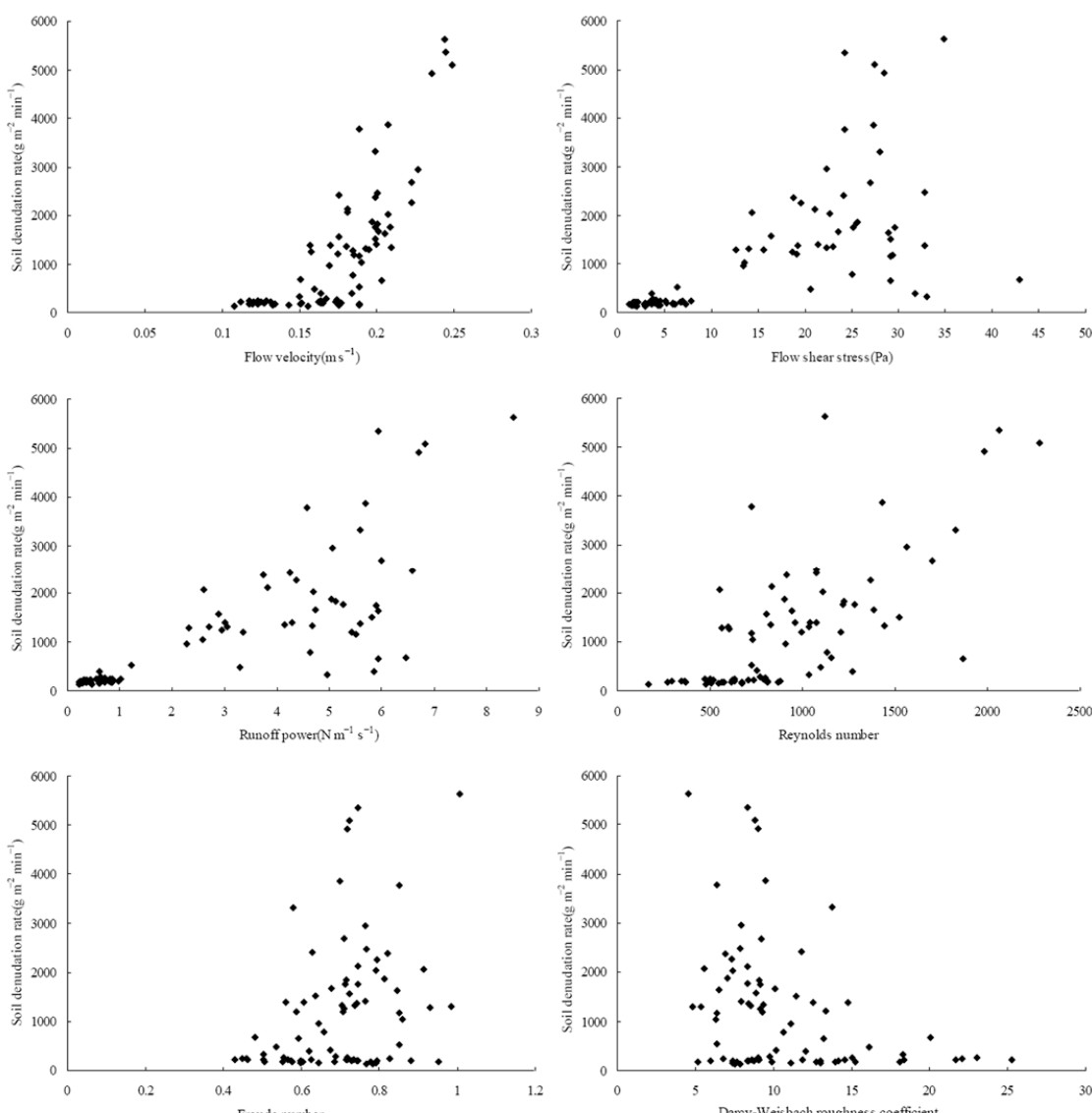

**Figure 8.** Relation between soil denudation rate and hydraulic parameters.

**Table 2.** Pearson correlation coefficients (*R*) between soil denudation rate and hydraulic parameters.

| Plots | Parameter | Flow Velocity | Flow Shear Stress | Runoff Power | Reynolds Number | Froude Number | Darcy–Weisbach Roughness Coefficient |
|---|---|---|---|---|---|---|---|
| GF1 | *R* | −0.243 | 0.427 | 0.330 | 0.496 [b] | −0.631 [a] | 0.680 [a] |
| | *p* | 0.301 | 0.061 | 0.155 | 0.026 | 0.003 | 0.001 |
| | *n* | 20 | 20 | 20 | 20 | 20 | 20 |
| GF2 | *R* | 0.792 [a] | 0.984 [a] | 0.988 [a] | 0.447 [b] | 0.596 [a] | −0.485 [b] |
| | *p* | 0.000 | 0.000 | 0.000 | 0.048 | 0.006 | 0.030 |
| | *n* | 20 | 20 | 20 | 20 | 20 | 20 |

**Table 2.** *Cont.*

| Plots | Parameter | Flow Velocity | Flow Shear Stress | Runoff Power | Reynolds Number | Froude Number | Darcy–Weisbach Roughness Coefficient |
|---|---|---|---|---|---|---|---|
| GF3 | $R$ | 0.639 [a] | -0.206 | 0.060 | 0.026 | 0.438 | −0.560 [b] |
| | $p$ | 0.002 | 0.384 | 0.801 | 0.915 | 0.054 | 0.010 |
| | $n$ | 20 | 20 | 20 | 20 | 20 | 20 |
| GF4 | $R$ | 0.774 [a] | 0.423 | 0.654 [a] | 0.701 [a] | 0.373 | −0.429 |
| | $p$ | 0.000 | 0.063 | 0.002 | 0.001 | 0.106 | 0.059 |
| | $n$ | 20 | 20 | 20 | 20 | 20 | 20 |
| All the fissure | $R$ | 0.764 [a] | 0.659 [a] | 0.771 [a] | 0.709 [a] | 0.327 [a] | −0.326 [a] |
| | $p$ | 0.000 | 0.000 | 0.000 | 0.000 | 0.003 | 0.003 |
| | $n$ | 80 | 80 | 80 | 80 | 80 | 80 |

Notes: [a] Correlation is significant at the 0.01 level (2-tailed). [b] Correlation is significant at the 0.05 level (2-tailed). *N* is the sample number.

**Table 3.** Relationship between soil denudation rate and hydraulic parameters.

| Flow Velocity | Flow Shear Stress | Runoff Power | Reynolds Number | Froude Number | Darcy-Weisbach Roughness Coefficient |
|---|---|---|---|---|---|
| 0.781 | 0.851 | 0.866 | 0.801 | 0.763 | 0.716 |

## 4. Discussion

### 4.1. Influence of Wetting Front on Soil Erosion

Rainfall infiltration directly affects the distribution characteristics of soil moisture and soil erosion on the slope [35–37]. Ground fissures are a typical geological hazard in coal mining areas [38]. The movement path of surface runoff on the platform was changed by ground fissures, leading to the water flow changing from a horizontal flow to longitudinal movement [29]. The soil infiltration was changed because a large amount of rainwater and surface runoff flowed into the ground fissure [39,40]. This also increased the possibility of soil and water loss disasters [13,22]. Hence, analyzing the role of ground fissures on infiltration, wetting front, runoff, and sediment in a dump was key to reveal the mechanisms of slope stability and soil erosion.

The soil on the slope had a high infiltration rate at the beginning of the test, and all rainfall infiltrated into the soil, leading to a downward movement of the wetting front. Guebert and Gardner [41] found that ground fissures provided preferential channels for water movement, resulting in more water moving to the deep soil. At the same time, the runoff collected by the platform of the dump flowed into the ground fissure. The volume of ground fissure was determined by its size, such as length, width, and depth, which affected the amount of stored rainwater. In our study, although the platform area of the dump was same, the depth of ground fissure was different for each test. Therefore, the deeper the ground fissures, the greater the stored rainwater. It indicated that ground fissures affected the infiltration process of the rainwater (Figure 3). The results were consistent with the results from Fu et al. [42], who found that ground fissures had a significant impact on near-surface hydrological processes. Furthermore, the deeper the ground fissures, the more obvious the water filling of the ground fissure was, the lower the depth of the wetting front to move. Relevant studies had shown that the shape characteristics and number of ground fissures had a significant influence on water movement [43,44]. We found that the wetting front moved deeper around ground fissures. This result was consistent with Zhang et al. [22]. Furthermore, the wetting front was relatively close at the beginning of the test. With the rainfall time increasing, the difference of wetting front became more and more obvious, and the difference reached the maximum after the end of the test. However,

this was related to the rainfall time, and if the rainfall was long enough, the difference between wetting fronts decreased.

### 4.2. Effect of Ground Fissure on Erosion Failure

The dump, with the compact rock platform and high and steep loose slope, was formed during coal mining, and is the most serious area of soil erosion in a surface coal mine [10]. The confluence of a compact platform was one of the most impactful causes of soil erosion. Relevant studies have shown that the slope in a dump was the main source of soil erosion, accounting for about 85% of the erosion [15,45]. Zhang et al. [22] found that slope failure could be divided into three stages. In our study, the erosion failure mode could be divided into the surface erosion stage, fissure deformation stage, rill erosion stage, fissure collapse-rapid increase stage, and stable stage. There was a difference on the type and characteristics of soil erosion within different stages. Su et al. [28] found that small runoff erosion could also induce natural disasters in a dump such as shallow landslides and collapses. Shallow mudflow may occur in the dump because runoff would come out from the slope toe through a seepage channel [46].

Relevant studies have shown that the ground fissure had a significant influence on soil erosion [43,44]. Soil erosion failure mode in GF2 was different from other conditions (Figure 9). At the beginning of the test, runoff not only flowed directly from the slope, but also flowed into the ground fissure and filled it. With the rainfall time increasing, the soil water content increased and tended to a saturated state. Infiltration of rainwater increased soil bulk density, and the soil on the slope became wet and soft. This may be responsible for soil erosion on the slope. Pu et al. [47] also found that the mechanical properties of soil on the slope would decrease sharply under rainfall conditions.

In our study, Figures 5–7 indicate that the runoff rate, erosion rate, and soil denudation rate at 6 min increased rapidly because of shallow mudflow, a gravitational erosion type, similar to the low gravitational downslope movement of water-saturated soil [48]. The shallow mudflow induced by rainfall was one of the forms of soil slope instability [49]. At this stage, the soil on the slope had high water content, high sand content, strong fluidity, and certain viscosity (Figure 9). Although the occurrence of the shallow mudflow had a certain randomness and happened in a short time, the soil erosion rate was huge [50]. Once the shallow mudflow was over, soil erosion significantly decreased. Our results have shown that the soil erosion accounted for 36.96% of the total erosion amount, which was 6.39 times higher than the average erosion amount (Figure 6). The destruction of the force balance state of the soil led to the change of the micro-topography on the slope, forming a free surface. The fissure, appearing above the shallow mudflow area on the slope, developed more rapidly because of creep deformation. Then, the shallow mudflow occurred again under the action of fissure development and the soil on the slope slid down the slope surface, until it reached the front edge of the slope top. However, compared with 6 min, the soil erosion rate caused by shallow mudflow decreased in the following period. The soil at the slope shoulder (connection between the platform and the slope) was eroded or even collapsed under the combined action of hydrostatic pressure in the ground fissure and runoff on the slope. Fissures at the slope shoulder were the key factor controlling the slope failure [23]. This may be responsible for the increase of erosion rate and soil denudation rate in the later period of the rainfall. However, it was difficult to determine the starting criterion of the shallow mudflow on the slope. At the same time, rill erosion continued to develop until it tended to stabilize. That is, the erosion gully expanded in width and cut down in depth. In summary, soil erosion in the dump with ground fissures was mainly shallow mudflow and rill erosion, and its result was the combined effect of hydraulic erosion and gravity erosion. Therefore, controlling the development of ground fissures and strengthening the drainage of platforms were the keys to prevent soil erosion of the dump in a surface coal mine.

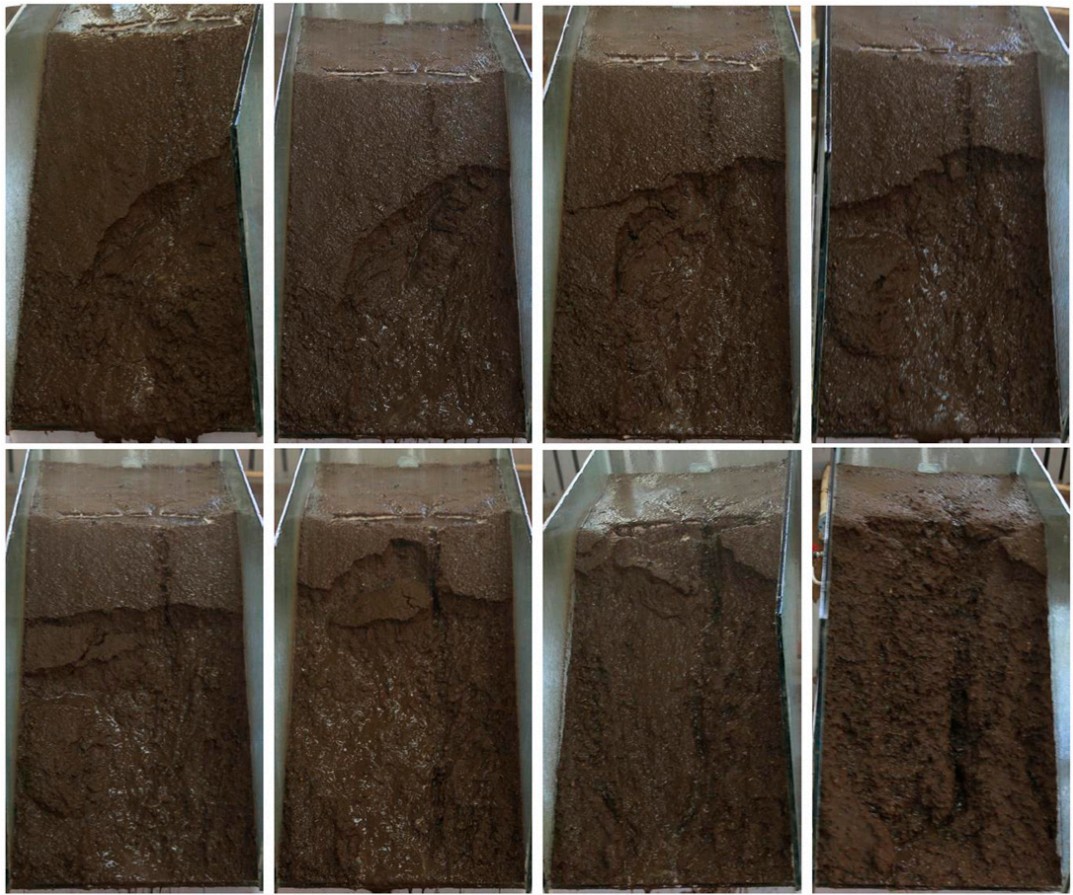

**Figure 9.** Soil erosion failure modes in GF2.

## 5. Conclusions

The wetting front depth with different fissure depths was in the order GF4 > GF3 > GF2 > GF1, indicating that ground fissure had influence on the movement of the wetting front. Erosion rates presented successive fluctuations with multiple peaks and valleys. In addition, the ground fissure depth had an impact on soil erosion. The correlation between soil denudation rate and runoff power was the highest. In our study, the erosion failure modes with platform-slope system in the dump could be divided into the surface erosion stage, fissure deformation stage, rill erosion stage, fissure collapse-rapid increase stage, and stable stage. Soil erosion in the dump with ground fissures was mainly shallow mudflow and rill erosion, and its result was the combined effect of hydraulic erosion and gravity erosion. Therefore, controlling the development of ground fissures and strengthening the drainage of platforms were the keys to prevent soil erosion of the dump in a surface coal mine.

**Author Contributions:** Conceptualization, Y.L. and G.L.; methodology, Y.L. and G.L.; software, Y.L. and G.L.; validation, W.S.; formal analysis, Y.L. and G.L.; investigation, Y.L.; resources, G.L.; data curation, Y.L.; writing—original draft preparation, Y.L. and G.L.; writing—review and editing, Y.L. and G.L.; visualization, D.W.; supervision, Z.W.; project administration, Y.L. and G.L.; funding acquisition, Y.L. and G.L. All authors have read and agreed to the published version of the manuscript.

**Funding:** This research was jointly funded by the Liaoning Province "Xing Liao Talents Program" Project (Grant. No. XLYC2007046), the National Key Research and Development Plan Subject (Grant. No. 2017YFC1503105), the Engineering and Technology Double First-Class Discipline Innovation Team Construction Project of Liaoning Technical University (Grant. No. LNTU20TD-24), and the Youth Teacher Training Fund Project of Shenyang University of Technology (Grant. No. 200005781).

**Data Availability Statement:** The data presented in this study are available on request from the corresponding author.

**Conflicts of Interest:** The authors declare no conflict of interest.

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
