# Peer review of "Erosion Failure of Slope in a Dump with Ground Fissure under Heavy Rain"

_water, doi:10.3390/w14213425_

Round 1

Reviewer 1 Report

In this manuscript, the authors investigated the erosion failure of slope under rainfall. The data were well analyzed, the contents of this paper are full and complete, the structure is reasonable. The overall language of the manuscript is Ok. Language issues have been observed regarding the correct usage of articles and abbreviations e.g. “GF” in line 95-96. All the Figures and Tables are relevant and well presented.

The experimental design is easy to follow and the study can be helpful to understand the soil erosion in a dump. The study site description needs more details and it must be revised. In the study site description, the climate conditions have been described in detail but it's not clear what is the main source of this information, how many years data have been used to get these averages?.

The results are well presented and support the conclusions. The discussion is sound and relevant, the literature cited in the introduction is nicely used for discussion. The results will be more meaningful if Authors further elaborate regarding: Figure 7. in Line 202, there was only 4 data and the regression equation was unnecessary. The soil denudation rate at 6 min in figure 8, in line 214, was higer than others, why? How to define time? The preparation stage of the test included rainfall?

Comments for the authors

The number of data for regression equation need to be supplemented, e.g. line 125-128, 164-167, 182-185.

L57-67: The whole introduction failed to introduce recent research progress on soil erosion with fissure. I think this part should be thoroughly enhanced.

L89-93: The soil bulk density of lower layer and upper layer were 1.7 and 1.35 g/cm3, why? How to determine?

L294-301: I still cannot understand why the erosion failure mode for GF2 was different with others. Please further clarify.

Author Response

Response to Reviewer 1 Comments

Point 1: In this manuscript, the authors investigated the erosion failure of slope under rainfall. The data were well analyzed, the contents of this paper are full and complete, the structure is reasonable. The overall language of the manuscript is Ok. Language issues have been observed regarding the correct usage of articles and abbreviations e.g. “GF” in line 95-96. All the Figures and Tables are relevant and well presented.

Response 1: Thank you for this suggestion. I have added abbreviations where ground fissures first appear.

Point 2: The experimental design is easy to follow and the study can be helpful to understand the soil erosion in a dump. The study site description needs more details and it must be revised. In the study site description, the climate conditions have been described in detail but it's not clear what is the main source of this information, how many years data have been used to get these averages?.

Response 2: Thank you for this suggestion. The climate conditions are mainly derived from China Meteorological science data sharing service platform, with an average value of 30 years.

Point 3: The results are well presented and support the conclusions. The discussion is sound and relevant, the literature cited in the introduction is nicely used for discussion. The results will be more meaningful if Authors further elaborate regarding: Figure 7. in Line 202, there was only 4 data and the regression equation was unnecessary. The soil denudation rate at 6 min in figure 8, in line 214, was higer than others, why? How to define time? The preparation stage of the test included rainfall?

Response 3: Thank you for this suggestion. Figure 7 had been deleted. Soil denudation rate at 6 min in Figure increased rapidly because of shallow mudflow and at this stage, the soil on the slope had high water content, high sand content, strong fluidity and certain viscosity. We explained in detail in the discussion. The outlet is set at the front edge of the soil trough, which is the runoff volume. After the occurrence of slope runoff, the time of runoff was recorded. And then, the runoff sediment samples were collected each 3 min and the rainfall lasts for 60 min.

Point 4: The number of data for regression equation need to be supplemented, e.g. line 125-128, 164-167, 182-185.

Response 4: Thank you for this suggestion. The number of data for regression equation had been supplemented.

Point 5: L57-67: The whole introduction failed to introduce recent research progress on soil erosion with fissure. I think this part should be thoroughly enhanced.

Response 5: Thank you for this suggestion. Relevant research progress was supplemented in introduction.

Point 6: L89-93: The soil bulk density of lower layer and upper layer were 1.7 and 1.35 g/cm3, why? How to determine?

Response 6: Thank you for this suggestion. Soil bulk density was determined by measuring soil samples which collected in the dump.

Point 7: L294-301: I still cannot understand why the erosion failure mode for GF2 was different with others. Please further clarify.

Response 7: Thank you for this suggestion. We explained in detail in the discussion.

Thanks again for your good comments on our paper very much.

If you have any question about this paper, please don’t hesitate to let me know.

Sincerely yours,

Prof. Gang Lv

Reviewer 2 Report

Dear authors,

It is an interesting paper that addresses an important topic. Soil erosion in active or closed coal mining areas causes a lot of problems ranging from the erosion itself, instability, etc. to rivers clogging and subsequent flooding events.

The result of your research are somehow as expected and confirm previous studies and theory.

The paper must be read carefully by a good English speaker, as some moderate corrections are needed (there are some mistakes and unfortunate expressions).

I really appreciate your effort and time consuming experiments, as well as the interpretations. However I do have a problem with lab experiments, especially when they involve physical models. Any physical model, no matter how hard we try, will never be able to replicate the natural conditions (it is impossible to add all the potential relevant variables in a lab model). Another problem is related to the equivalent material considered in the experiments (how close is this material to the actual mixture of rocks from the waste dump ?)

To conclude, I strongly believe that lab experiments should always be accompanied by field experiments/measurements/observations.

Despite the above comments, I will recommend the editors to publish your research after minor revisions (related to the overall quality of the English and some related to the format of the paper: caption of Table 2 must be on the next page, space between figure caption and text, space between text and Table 3, etc.).

Good luck and keep up the good work!

Author Response

Response to Reviewer 2 Comments

Point 1: I really appreciate your effort and time consuming experiments, as well as the interpretations. However I do have a problem with lab experiments, especially when they involve physical models. Any physical model, no matter how hard we try, will never be able to replicate the natural conditions (it is impossible to add all the potential relevant variables in a lab model). Another problem is related to the equivalent material considered in the experiments (how close is this material to the actual mixture of rocks from the waste dump ?)

Response 1: Thank you for this suggestion. When conducting indoor simulation research, the similarity principle was usually used to determine the geometric size and material of the dump. According to the field survey data, such as the length, width and height of the dump, combined with the indoor simulation test, the similarity coefficient was determined to be 50 in this study. Coal gangue with particle size of 10-30 mm and soil with particle size of 0-10 mm were used as test materials. Of course, we had tried to be as close to natural conditions as possible.

Point 2: caption of Table 2 must be on the next page, space between figure caption and text, space between text and Table 3, etc.).

Response 2: Thank you for this suggestion. We had made corresponding corrections.

Thanks again for your good comments on our paper very much.

If you have any question about this paper, please don’t hesitate to let me know.

Sincerely yours,

Prof. Gang Lv
